# Improving the Efficiency of Distributed Training using Sparse Parameter Averaging

**Matt Beton**[1]  **Seth Howes**[1]  **Alex Cheema**[1]  **Mohamed Baioumy**[1]  **Matt Reed**[2]

[1]Exo Labs  [2]Stanford University

## Abstract

Large language model (LLM) training is typically distributed across many accelerators to reduce training time, necessitating frequent exchange of information across high-speed, low-latency networks. Federated learning algorithms like DiLoCo have relaxed this requirement by grouping accelerators into islands, between which communication is infrequent. In the case of DiLoCo, synchronization between workers happens every $H$ steps, thus reducing the communication cost by a factor of $H$. However, if $H$ is too large, model convergence is affected as nodes performing local optimization diverge too far. In this work, we explore Sparse Parameter Averaging (referred to as SPARTA), where models asynchronously share a small subset of the parameters (e.g., 0.05%) at each training iteration. This keeps them within the same basin to reduce divergence between models. The main contribution of this paper is to combine SPARTA with DiLoCo, which provides two benefits over 'pure' DiLoCo. First, using SPARTA increases correlation between nodes. This enables a 100× increase in the DiLoCo interval without incurring additional wall-clock time, whilst still achieving performance gains. Second, we show that SPARTA acts as a regularizer, allowing for a higher learning rate and faster convergence.

## 1 Introduction

Training large language models (LLMs) such as GPT (Radford et al., 2019) and BERT (Devlin, 2018) requires immense computational resources, often distributed across many accelerators. While this parallelism reduces training time, it introduces substantial communication overhead, as model parameters and gradients must be synchronized frequently between devices. This synchronization typically requires high-bandwidth, low-latency interconnects, making large-scale training infeasible outside well-connected data centers.

Advancements in federated learning algorithms relax these communication requirements. A popular algorithm for this is FedOpt (Reddi et al., 2020), which groups accelerators into loosely connected "workers" that independently take multiple local optimization steps (inner optimization), before synchronizing their parameters across workers every $H$ steps (outer optimization). This reduces communication costs by a factor of $H$, allowing for training across devices with limited interconnectivity. A recent variation of this approach is Distributed Low-Communication (DiLoCo) (Douillard et al., 2024), which uses AdamW for the inner optimizer, and Nesterov momentum for the outer optimization step. Critically, DiLoCo's performance can match data parallelism approaches (full synchronization at every step), which was not the case with previous variations like FedOpt.

More recent approaches to further reduce bandwidth requirements have been introduced by Jaghouar et al. (2024) and Douillard et al. (2025) who split synchronizations over several timesteps as opposed to a single timestep, as well as quantizing the data exchanged across workers. In both papers, the model synchronizations were executed every 100 steps, equivalent to a 100× reduction in communication.

Increasing H to larger value would be very beneficial, as it further reduces communication overhead, therefore reducing the bandwidth required for distributed training to be feasible. However, for pure DiLoCo with $H > 100$, the lack of synchronization causes local workers to diverge, reducing overall performance.

Correspondence: `matt@exolabs.net`

In this work, we explore Sparse Parameter Averaging (SPARTA) as a promising solution to reduce communication during distributed training. At each step SPARTA averages a small subset of model parameters across workers, maintaining alignment between workers while significantly lowering communication costs. We find that SPARTA communications can be carried out asynchronously, thus not blocking workers from training. A parameter from a previous time-step is shared thus overlapping training at time step $t$ and parameter sharing from the model at $t - 1$. Thus SPARTA does not incur any additional wall-clock time.

We propose combining DiLoCo and SPARTA to address high communication overhead and convergence degradation at high $H$. By asynchronously sharing a sparse subset of model parameters during local training, our approach maintains convergence while reducing frequency of full synchronizations (DiLoCo synchronizations).

We show that our method achieves two key improvements. First, combining DiLoCo with SPARTA induces a regularizing effect that allows the use of a higher learning rate on local nodes, thus allowing models to perform better in fewer time-steps. Second, SPARTA induces model populations to stay in the same loss basin. In our experiments, we show that we can reduce the synchronous DiLoCo communication frequency $H$ from every 100 steps to every 10,000 steps whilst achieving a lower validation perplexity.

Our contributions demonstrate that integrating sparse communication with infrequent synchronization makes large-scale distributed training more viable, as the frequency of synchronous 'breaks' for workers to synchronize weights is reduced by $100\times$, providing better Model FLOPs Utilization (MFU) in a low-bandwidth environment.

## 2 METHOD

We assume a base model architecture (e.g. a transformer) with parameters $\theta$. We denote a training dataset as $\mathcal{D} = \{(x, y), \ldots\}$. We consider the standard language modeling task, where the objective is to predict the next token in a sequence given all previous tokens.

In our experiments, we instantiate this formulation using a 124M parameter version of nano-GPT, a minimal yet capable transformer-based language model. The nano-GPT architecture closely follows the design of GPT-2 small (Radford et al., 2019), but has a minimal implementation that enables rapid experimentation. The model comprises 12 transformer blocks, each containing 12 self-attention heads, with an embedding dimension of 768. This configuration offers a balanced trade-off between expressiveness and computational efficiency, meaning we can efficiently assess performance across multiple nodes whilst ensuring generalizability to models with larger parameter counts.

We consider data-parallel techniques, where workers contain full copies of the model, and compute forward- and backward-passes on their local dataset. We split dataset $\mathcal{D}$ homogeneously across a variable number of workers, with the $i$-th shard denoted as $\mathcal{D}_i$. In standard data-parallel training, gradients are synchronized and averaged between workers at every step. This ensures that each accelerator takes a step in parameter space based on the global consensus of gradients. However, this synchronous full-model AllReduce is communication-intensive, and requires all workers to pause training during synchronization.

### 2.1 OVERVIEW OF DiLoCo

Distributed Low-Communication Optimization (DiLoCo) (Douillard et al., 2024) addresses the challenge of communication overhead in large-scale distributed training by reducing the frequency of full synchronization across worker nodes.

For each DiLoCo interval, each worker initializes a local copy of the model. Workers then independently compute local updates using AdamW (Loshchilov & Hutter, 2019) for $H$ inner steps on a given subset of the training data $D_i$. Every $H$ optimization steps, all workers synchronize using stochastic gradient descent (SGD) with Nesterov momentum (Nesterov, 1983), which updates parameters using a velocity term, where the gradient is computed at a predicted future position $\theta - \gamma v$ rather than the current position $\theta$.

This process reduces communication overhead by a factor of $H$, making it suitable for distributed environments with limited network bandwidth.

## 2.2 SPARTA: SPARSE PARAMETER AVERAGING

We are inspired by WASH (Fournier et al., 2024), in which a random subset of weights is picked at each training step, and weights are permuted between workers. In Sparse Parameter Averaging for Reduced communication TrAining (SPARTA), a subset of model parameters are instead averaged across workers. At each local timestep, a subset of the model weights $\theta_s \subset \theta$ is selected, with $|\theta_s| = p|\theta|$. We call $p$ the 'SPARTA parameter' - the proportion of weights that are communicated at each timestep. This subset of parameters is communicated among workers and each weight is updated to be the global average of that parameter across workers. SPARTA can be used as a standalone method for distributed training (figure 4), however we will later combine this with the DiLoCo algorithm.

In SPARTA parameters are shared from a previous timestep thus allowing for overlapping communication and training, and incurs no additional wall-clock time. The asynchronous property of SPARTA is possible due to the algorithm averaging model *weights*, instead of gradients (as most distributed training algorithms do). Model gradients can change drastically between training steps, while model weights are relatively much slower moving. This shows the power of SPARTA as an asynchronous algorithm: for modern language models of 100B+ parameters, training iterations will typically take multiple minutes per iteration (BigScience Workshop, 2022). This shows that we can have a large communication delay due to network issues without any performance degradation.

## 2.3 DiLoCo + SPARTA

While DiLoCo effectively reduces communication overhead by synchronizing parameters infrequently, large synchronization intervals ($H$) can allow local models to diverge too far from each other, leading to a large reduction in training performance. Model synchronization steps lead to a large spike in perplexity due to models having drifted too far (figure 5 in the appendix).

To mitigate this, we propose integrating SPARTA with DiLoCo. SPARTA is able to maintain model alignment without any full-synchronization steps. We thus leverage SPARTA to improve performance of DiLoCo by reducing model divergence between full-sync steps.

Combining SPARTA with DiLoCo affords the following improvements to distributed training. Firstly, it reduces the need for frequent synchronizations between workers. By employing sparse communication local models remain within a similar optimization basin, reducing the negative effects of large $H$. Our methods allow for increased $H$ values (e.g., from 100 to 10,000 steps) without performance degradation, making distributed training more feasible across resource-constrained or decentralized environments as synchronous updates are applied much less frequently. Secondly, sparse updates have a regularizing effect on local models, enabling the use of higher learning rates, which accelerates convergence.

## 3 RESULTS

### 3.1 REGULARIZING EFFECT OF SPARTA

We found that DiLoCo + SPARTA improves model performance for a fixed number of optimization steps (Figure 2). Furthermore, we noticed that tuning the learning rate with DiLoCo + SPARTA resulted in a much higher optimal LR. The learning rate was increased by a factor of 1.75 whilst still maintaining training stability, which we believe is an introduced regularization effect as a result of the randomness introduced by SPARTA updates from other models in the population.

### 3.2 MODEL CORRELATIONS

We find that when using DiLoCo to train models distributed across nodes, the average pairwise correlation (calculated as shown in A.3) between worker model parameters progressively decreases over a DiLoCo interval (Figure 1). However, through combining DiLoCo with SPARTA, the average

---

**Algorithm 1** DiLoCo with Sparse Parameter Averaging (SPARTA)

---

1: **Require:** Initial model $\theta^{(0)}$
2: **Require:** $k$ workers
3: **Require:** Data shards $\{D_1, \ldots, D_k\}$
4: **Require:** Inner optimizer `AdamW`
5: **Require:** Outer optimizer `DiLoCo`
6: **Require:** Synchronization interval $H$ and sparsity fraction $p$ (e.g., $p = 0.0005$)
7: **for** step $t = 1, \ldots, T$ **do**
8:     $\mathcal{P}^{(t)} \subseteq \theta_i^{(t)}$ indices randomly sampled, with $|\mathcal{P}^{(t)}| = p|\theta^{(t)}|$
9:     **for** worker $i = 1, \ldots, k$ **in parallel do**
10:         $\theta_i^{(t)} \leftarrow \texttt{AdamW}(\theta_i^{(t-1)}, D_i)$
11:         Asynchronously communicate $\mathcal{P}_i^{(t)} \subset \theta_i^{(t)}$ with other workers
12:         Update $\theta_i^{(t)}$ with $\frac{1}{k} \sum_i \mathcal{P}_i^{(t)}$ for parameters to be updated
13:         **if** $t \bmod H == 0$ **then**
14:             Communicate model parameters $\{\theta_j^{(t)}\}_j$
15:             $\theta_i^{(t)} \leftarrow \texttt{DiLoCo}\left(\theta^{(t-1)}, \{\theta_j^{(t)}\}_j\right)$
16:         **end if**
17:     **end for**
18: **end for**=0

---

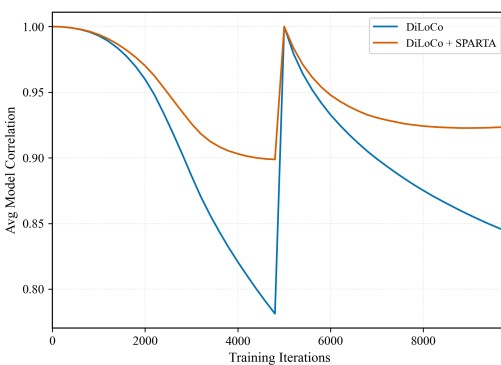

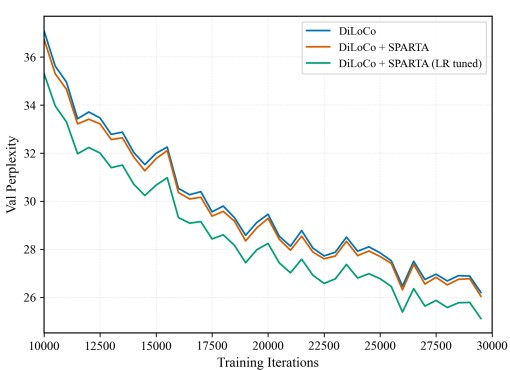

Figure 1: When using DiLoCo + SPARTA, model correlation is higher than pure DiLoCo.

Figure 2: DiLoCo with $H = 100$. SPARTA provides a regularizing effect, allowing for higher learning rates.

pairwise correlation between model parameters is higher when compared to DiLoCo alone. At the end of the first DiLoCo interval ($H = 1000$), models trained using DiLoCo alone had an average pairwise correlation of 78.1%, demonstrating large divergence between worker models. However, the average pairwise correlation for models trained using DiLoCo + SPARTA with $p = 0.05\%$ was notably higher at 90.3%.

### 3.3 INCREASED DILOCO INTERVAL $H$

As observed in Douillard et al. (2024), we find that training performance significantly degrades when $H$ is increased beyond 2000 steps. For higher $H$ models are allowed to diverge further, and when models are subsequently merged we see a reduction in performance (figure 5 in the appendix). However, by combining DiLoCo + SPARTA, we achieved a stronger training performance even with a $100\times$ increase in the DiLoCo interval (figure 3). For example, we observed a 14.3% decrease in model perplexity for $H = 10000$ when comparing DiLoCo alone to DiLoCo + SPARTA ($p = 0.5\%$) with a 2x learning rate increase.

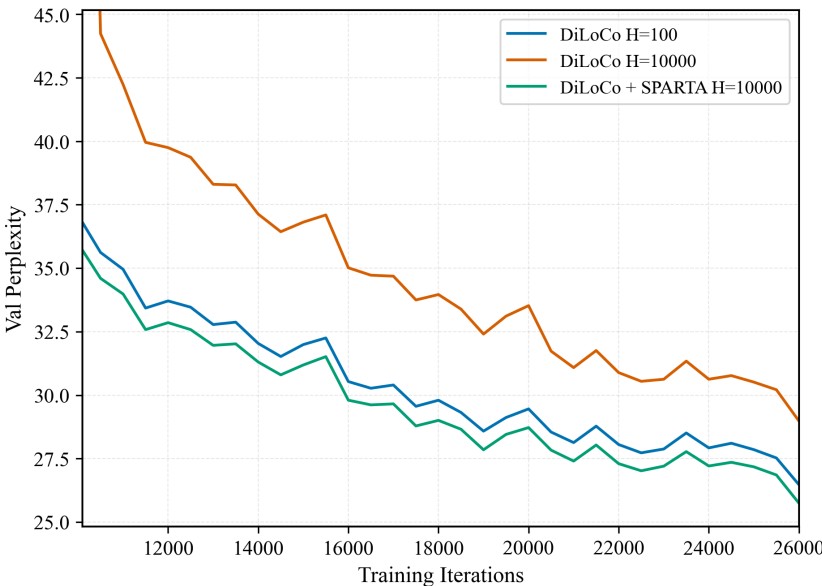

Figure 3: DiLoCo + SPARTA improves performance whilst reducing wall-clock time.

### 3.4  PARAMETER SHARING SCHEDULE

We find that linearly increasing the proportion of parameters shared between models at each training step from 0 to a maximum value of $2p$ every $H$ steps increases model correlation compared with a fixed $p$ schedule (figure 7 in the appendix). We find that despite this increased model correlation over the training run, there is no noticeable difference in the training performance relative to a fixed $p$-schedule. Our experiments imply that maintaining near-perfect model correlation throughout the training run is not necessarily the best strategy; some level of divergence may actually be beneficial. This opens up an interesting avenue for future research, where measuring and controlling the degree of divergence—or leveraging heterogeneous datasets—could help in identifying the optimal trade-off for effective distributed training.

## 4  RELATED WORK

The field of distributed training has been explored through a variety of methods in recent years. Methods typically aim to allow nodes to train independently with small inter-node communications that ensure models stay in the same loss basin.

Model Soups (Wortsman et al., 2022b), BTM (Li et al., 2022) and Lo-fi (Wortsman et al., 2022a) allow nodes to independently fine-tune a pretrained model, before parameter-averaging at the end of a training run. In our research, we noted that a population of models diverging too far can cause large performance drops when averaging models. Research has been done into alternative model-merging techniques. Yadav et al. (2023) seeks to reduce the performance drop by resolving conflicts on an individual weight basis. Yadav et al. (2024) analyses the effect of model size and start point on merging.

DART (Jain et al., 2023) and Federated Averaging (McMahan et al., 2023) periodically average model weights every $H$ local training steps. PAPA (Jolicoeur-Martineau et al., 2024) does not fully average models, but instead periodically pushes local weights slightly toward the population average. DiLoCo Douillard et al. (2024) furthered the idea of DART and PAPA by using a Nesterov Momentum (Nesterov, 1983) update each $H$ local steps.

Recent works have extended the capabilities of DiLoCo to improve distributed training. OpenDiLoCo (Jaghouar et al., 2024) used DiLoCo to train a billion-parameter model across continents. Streaming DiLoCo (Douillard et al., 2025) showed that peak bandwidth can be reduced by

synchronizing subsets, quantizing shared parameters, and allowing nodes to continue training whilst communicating weights. Asynchronous Local-SGD (Liu et al., 2024) developed an asynchronous Federated Averaging, at the expense of taking more training iterations to converge.

Finally, 'Weight Shuffling, then Average' (WASH) (Fournier et al., 2024) employs sparse parameter shuffling; picking a random subset of parameters at each training interval and *permuting* them between population models. WASH provided the inspiration for the SPARTA algorithm.

## 5 DISCUSSION AND FUTURE WORK

There are several promising directions for future research to improve DiLoCo + SPARTA. One natural extension is to explore optimizer selection beyond the currently used AdamW. Although AdamW is widely regarded as the optimizer of choice for training transformer-based language models, its suitability in the presence of SPARTA remains an open question. Future works could also investigate the application of SPARTA to models other than generative language transformer models.

Another direction involves experimenting with more flexible synchronization intervals and adapting DiLoCo scheduling. Our current approach uses a fixed synchronization interval of $H$ steps, however employing hybrid schemes to vary synchronization frequency throughout training could improve both convergence and scalability. Similarly, the SPARTA probability $p$ can be varied throughout the training run. As model population discorrelation reduces over the course of training, it may be advantageous to decrease the SPARTA probability $p$ or dynamically adjust the shuffling rate to better accommodate different phases of training.

In the setting of pure SPARTA, at each timestep a uniformly random proportion $p$ of parameters is communicated. After $n$ steps, the proportion of parameters that have been communicated at least once is:

$$1 - (1 - p)^n.$$

This equation is nonlinear in $p$. This can be seen numerically; for $p = 0.05\%, n = 10,000$ the proportion of parameters shared at least once is 99.3%, and by doubling for $p = 0.1\%, n = 10,000$ we have 99.996% shared. This firstly helps to explain why we have strong performance at low $p$-values, with sublinear improvements for increased $p$. Secondly, this also justifies the potential use of $p$-scheduling in further experiments.

We intend to investigate diverse sharing schedules and study the impact of varying latency between sending and incorporating model parameters. These factors could have a significant effect on both convergence behavior and overall system efficiency, offering further opportunities to optimize DiLoCo + SPARTA for large-scale distributed training.

Alternative asynchronous strategies also merit further study. While our current implementation of SPARTA employs asynchronous parameter exchange, future studies will evaluate different communication protocols that could further reduce overhead and improve system robustness. In this context, investigating gossip-based communication is particularly promising. Our current AllReduce-style parameter averaging incurs an $\mathcal{O}(n^2)$ communication scaling with the number of nodes. By contrast, gossip-based protocols (Giaretta & Girdzijauskas, 2019) have the potential to reduce this complexity to $\mathcal{O}(n)$, enabling efficient scaling to extremely large distributed settings.

## 6 CONCLUSION

In this paper we presented SPARTA as a powerful method for reducing communication overhead during distributed training. Further, SPARTA is shown to enhance the performance of DiLoCo when the synchronization interval is extended from 100 to 10,000 steps while communicating only 0.05% of parameters per iteration. This allows local nodes to adopt up to a 2× higher learning rate, yielding a 14.3% reduction in validation perplexity. By asynchronously exchanging a small subset of parameters, DiLoCo + SPARTA maintains strong model alignment despite infrequent full synchronizations, improving on the performance of DiLoCo with 100× less communication overhead.

Overall, DiLoCo + SPARTA presents a promising avenue for research of large-scale training in bandwidth-constrained settings. We hope this provides a starting point for further exploration of re-

lated distributed training strategies, as SPARTA is a generalizable technique that could be leveraged in a variety of distributed training settings.

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

# A APPENDIX

## A.1 STANDALONE SPARTA

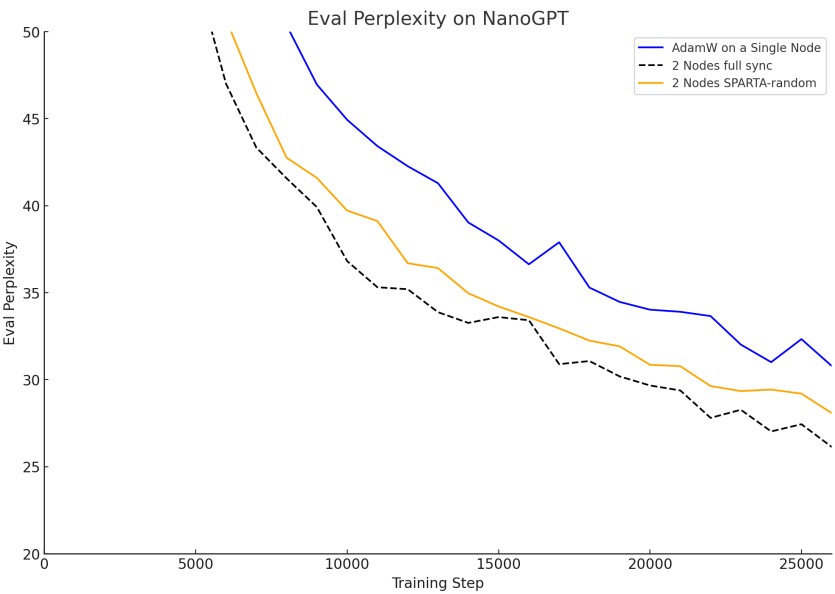

Figure 4: Pure SPARTA is able to perform comparatively to 2 nodes full-sync.

## A.2 MERGING OF HIGHLY DECORRELATED MODELS

## A.3 CALCULATING AVERAGE PAIRWISE MODEL CORRELATION

In this section, we demonstrate how pairwise correlation between models is calculated.

Given a population of models $\{\theta^{(i)}\}$, we flatten each model $\theta^{(i)}$ into a vector of weights $\mathbf{w}^{(i)} = \{w_k^{(i)}\}_k$. The Pearson correlation coefficient is then calculated between each pair of weight vectors, $\rho_{ij} := \rho(w^{(i)}, w^{(j)})$. The average pairwise correlation across all models is given by:

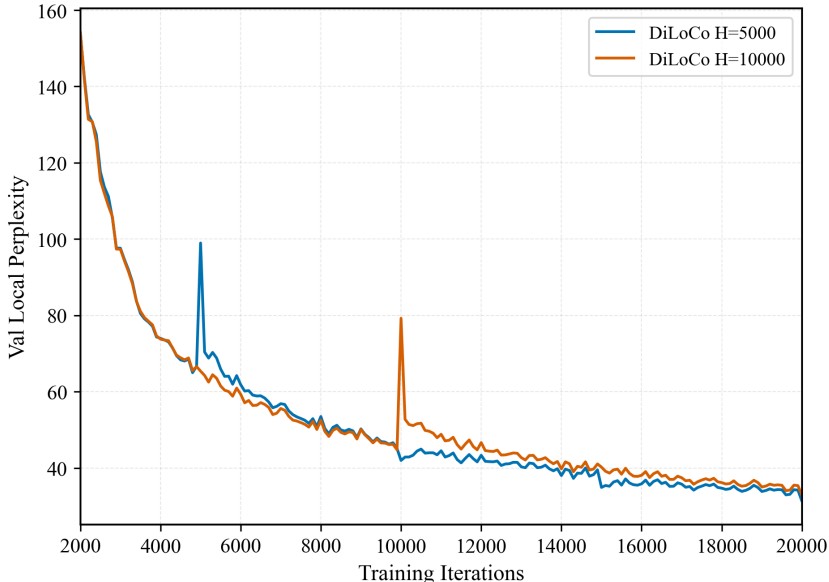

Figure 5: Pure DiLoCo with high $H$ leads to performance drops when averaging models due to low model correlation.

$$\bar{\rho} = \frac{2}{k(k-1)} \sum_{i<j} \rho_{ij}$$

## A.4    SCALED $p$-VALUES FOR SPARTA

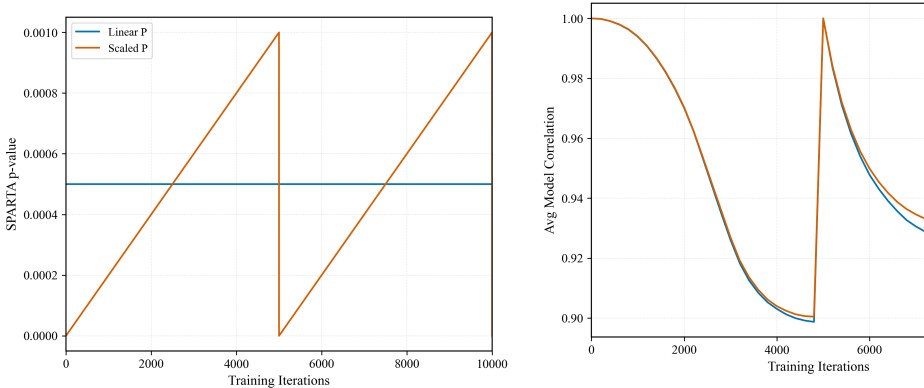

Figure 6: SPARTA $p$-value schedule.

Figure 7: Model correlation for linear and scheduled $p$.

