# OpenReview forum: "Improving the Efficiency of Distributed Training using Sparse Parameter Averaging"
_ICLR.cc/2025/Workshop/MCDC — MCDC @ ICLR 2025_

### Official Review · Reviewer_sf3z · 2025-02-24

**Rating:** 6
**Confidence:** 4
**Fit:** 5

**Summary:**

The paper introduces SPARTA (Sparse Parameter Averaging), a distributed training approach that asynchronously shares a tiny fraction (0.05%) of model parameters between workers at each training step. The authors combine this with DiLoCo, an existing distributed training method, to achieve two key benefits: 1) enabling much less frequent full model synchronization (every 10,000 steps vs 100 steps) without hurting performance, and 2) providing a regularization effect that allows higher learning rates. Using a 124M parameter nano-GPT model, they demonstrate a 14.3% reduction in validation perplexity while reducing communication overhead by 100x.

**Reason For Giving A Higher Score:**

The paper presents a practical and effective solution to a significant problem in distributed training. The 100x reduction in communication overhead while maintaining or improving performance is impressive. The approach is simple to implement and could have immediate impact for organizations training large models with limited networking infrastructure.

**Reason For Giving A Lower Score:**

The limited experimental validation and lack of theoretical analysis make it hard to fully trust the results will generalize. More rigorous comparisons against competing approaches and testing on larger models would strengthen the paper's claims. The regularization effect, while intriguing, needs better characterization.

**Strengths And Weaknesses:**

The proposed approach introduces a novel combination of sparse parameter sharing with DiLoCo that addresses a real pain point in distributed training and the authors show strong empirical results showing improved performance with drastically reduced communication. The paper is well-written and there is a clear explanation of how asynchronous parameter sharing helps maintain model alignment.

My concerns with the paper center mainly around the limited experimental validation; the authors only tested their approach on nano-GPT which is a relatively small model of 124M parameters so it's not clear if this approach will scale to larger models. I also would have liked to see more comparisons against other recent approaches like Async Local-SGD, as well as discussion of potential failure modes, limitations and how the approach scales with model size or number of workers.

**Suggestions:**

See concerns listed above.

---

### Official Review · Reviewer_mbfU · 2025-02-27

**Rating:** 7
**Confidence:** 3
**Fit:** 4

**Summary:**

This research paper tackles the significant challenge of reducing communication overhead in distributed training of large language models while maintaining their effectiveness. The authors present SPARTA (Sparse Parameter Averaging), a novel approach that works in conjunction with DiLoCo, a distributed low-communication optimization method. DiLoCo minimizes communication by synchronizing parameters less frequently. SPARTA complements this by asynchronously averaging a small subset of model parameters, effectively reducing communication without increasing processing time. When combined, these methods lead to better convergence when communication is less frequent. SPARTA also appears to function as a regularizer, allowing for faster convergence through higher learning rates.

**Reason For Giving A Higher Score:**

A good paper with a simple idea explained well.

**Reason For Giving A Lower Score:**

-

**Strengths And Weaknesses:**

SPARTA is a relatively simple idea that shows significant promise for improving distributed training with DiLoCo. The results shown are quite compelling (particularly Figure 3) and the reduction in inter-node communication is potentially quite significant when comparing H=100 versus H=10000, even when the cost of the per-iteration updates is included. The major omission in the paper is a fuller discussion of the overall performance in terms of wall-clock time. For example, the caption of Figure 3 states "whilst reducing wall-clock time" - by how much? I presume that the loss calculations dominate the overall wall clock time; does the reduction in inter-node communication provide a significant speed increase or is it relatively small because DiLoCo already improves speeds a lot? Even it if is small, there appears to be additional benefits to using this approach. While the paper states that the communication is asynchronous, have there been any experiments to determine how asynchronous it can be? Is it always assumed that the weight communication happens within one gradient update? Or can larger delays be tolerated? This would be interesting for situations where inter-node communication is particularly slow (e.g., where resources are not geographically co-located).

**Suggestions:**

See strengths and weaknesses.

A very minor point: it would be nice to include the pairwise correlation for DiLoCo with H=100 (page 3, line 158+).

Typos:
- "papaer" should be "paper".
- "communiaiton" should be "communication".
- "to additional wall-clock time" should be "no additional wall-clock time".
- Algorithm 1, line 7: "do do" should be "do"

---

### Official Review · Reviewer_aba5 · 2025-03-03

**Rating:** 6
**Confidence:** 4
**Fit:** 5

**Summary:**

The paper proposes averaging sets of sparse parameters across model replicas within the DiLoCo setting, effectively increasing the global synchronization interval with negligible information exchange overhead. Additionally, this approach appears to introduce a regularization effect, leading to improved model convergence.

**Reason For Giving A Higher Score:**

Reducing the communication overhead of DiLoCo is a valuable contribution, particularly since DiLoCo is a widely used baseline for decentralized data parallelism. The proposed approach can also be applied to other DiLoCo variants, enhancing its general applicability.

**Reason For Giving A Lower Score:**

N/A

**Strengths And Weaknesses:**

# Strengths

- Reducing the communication overhead of DiLoCo is a valuable contribution, particularly since DiLoCo is a widely used baseline for decentralized data parallelism. The proposed approach can also be applied to other DiLoCo variants, enhancing its general applicability.
- The regularization effect induced by sparse parameter averaging is intriguing, and its impact on convergence appears to be significant.
- The method increases model correlation, which can influence the final aggregated performance—a beneficial characteristic in many settings.

# Weaknesses

- The experiments were conducted on a relatively small model, making it unclear whether the observed characteristics would hold at scale. Additional experiments on larger models would strengthen the paper’s conclusions.
- The writing could be improved for clarity and readability (e.g., Line 050).
- The plots could be more polished and high-resolution to enhance professionalism and readability.

**Suggestions:**

Authors should consider adding more scaled up experiments. The current experimental setting can be quite unconvincing given the smaller size of the models.

It is not clear to me if there is a practical reduction of the wall clock time due to this approach since DiLoCo synchronization is already sparse. So other than the regularization effect, is there any practical advantage in the decentralized settings? If there is, please make it clearer.

---

### Decision · Program_Chairs · 2025-03-06

**Decision:**

Accept

**Comment:**

This work proposes to communicate only a sparse set of parameters at each training iteration when using local optimization algorithms such as diloco, while computing full averaging every H steps. Results are promising when increasing H, and reviewers recommend acceptance.  We encourage the authors to carefully consider the reviewers' comments and suggestions when preparing the final version.